# MULTI-GRID TENSORIZED FOURIER NEURAL OPERATOR FOR HIGH RESOLUTION PDES

## ABSTRACT

Memory complexity and data scarcity are two main pressing challenges in learning solution operators of partial differential equations (*PDE*) at high resolutions. These challenges limited prior neural operator models to low/mid-resolution problems rather than full scale real-world problems. Yet, these problems possess spatially local structures that is not used by previous approaches. We propose to exploit this natural structure of real-world phenomena to predict solutions locally and unite them into a global solution. Specifically, we introduce a neural operator that scales to large resolutions by leveraging local and global structures through decomposition of both the input domain and the operator's parameter space. It consists of a multi-grid tensorized neural operator (*MG-TFNO*), a new data efficient and highly parallelizable operator learning approach with reduced memory requirement and better generalization. *MG-TFNO* employs a novel multi-grid based domain decomposition approach to exploit the spatially local structure in the data. Using the *FNO* as a backbone, its parameters are represented in a high-order latent subspace of the Fourier domain, through a global tensor factorization, resulting in an extreme reduction in the number of parameters and improved generalization. In addition, the low-rank regularization it applies to the parameters enables efficient learning in low-data regimes, which is particularly relevant for solving *PDE*s where obtaining ground-truth predictions is extremely costly and samples, therefore, are limited. We empirically verify the efficiency of our method on the turbulent Navier-Stokes equations where we demonstrate superior performance, with 2.5 times lower error, $10\times$ compression of the model parameters, and $1.8\times$ compression of the input domain size. Our tensorization approach yields up to 400x reduction in the number of parameter without loss in accuracy. Similarly, our domain decomposition method gives a $7\times$ reduction in the domain size while slightly improving accuracy. Furthermore, our method can be trained with much fewer samples than previous approaches, outperforming the *FNO* when trained with just half the samples.

## 1 INTRODUCTION

Real-world scientific computing problems often time require repeatedly solving large-scale and high-resolution partial differential equations (*PDE*s). For instance, in weather forecasts, large systems of differential equations are solved to forecast the future state of the weather. Due to internal inherent and aleatoric uncertainties, multiple repeated runs are carried out by weather scientists every day to quantify prediction uncertainties. Conventional *PDE* solvers constitute the mainstream approach used to tackle such computational problems. However, these methods are known to be slow and memory-intensive. They require an immense amount of computing power, are unable to learn and adapt based on observed data, and often times require sophisticated tuning (Slingo & Palmer, 2011; Leutbecher & Palmer, 2008; Blanusa et al., 2022).

Neural operators are a new class of models that aim at tackling these challenging problems (Li et al., 2020b). They are mappings between function spaces whose trained models emulate the solution operators of *PDE*s (Kovachki et al., 2021b). In the context of *PDE*s, these deep learning models are orders of magnitude faster than conventional solvers, can easily learn from data, can incorporate physically relevant information, and recently enabled solving problems deemed to be unsolvable with the current state of available *PDE* methodologies (Liu et al., 2022; Li et al., 2021b). Among

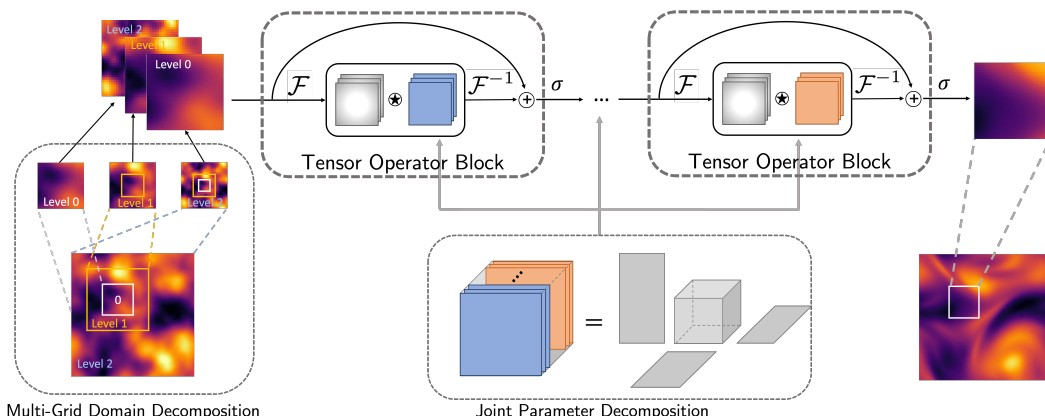

Figure 1: **Overview of our approach**. First (left), a multi-grid approach is used to create coarse to fine inputs that capture high-resolution details in a local region while still encoding global context. The resulting regions are fed to a tensorized Fourier operator (middle), the parameters of which are jointly represented in a single latent space via a low-rank tensor factorization (here, a Tucker form). Here $\mathcal{F}$ denotes Fourier transform. Finally, the outputs (right) are stitched back together to form the full result. Smoothness in the output is ensured via the choice of loss function.

| Method | $L^2$ test error | # Params | Model CR | Domain CR |
|---|---|---|---|---|
| **FNO** | 2.54% | 58 M | 0× | 0× |
| **TFNO** [Tucker] | 1.39% | 41 M | 1.5× | 0× |
| **TFNO** [CP] | 2.24% | 130 K | **482×** | 0× |
| **MG-FNO** | 1.43% | 58 M | 0× | 1.4× |
| **MG-TFNO** [Tucker] | **0.85**% | 5.5 M | 10× | 1.78× |
| **MG-TFNO** [Tucker] | 1.89% | 5.5 M | 10× | **7×** |

Table 1: **Overview of the performance on the relative $L^2$ test error of our *MG-TFNO* approach, compared with its parts *TFNO* and **MG-FNO** and the regular FNO, on Navier-Stokes.** CR stands for compression ratio. For each method, we report the relative $L^2$ error, the number of parameters, and the compression ratios for both the input domain and the number of parameters. Tensorization and multi-grid domain decomposition both individually improve performance while enabling space savings. The two techniques combined lead to further improvements, enabling huge compression for both input and parameter, while largely outperforming all other approaches.

neural operator models, Fourier neural operators (*FNO*s) in particular, have seen successful application in scientific computing for the task of learning the solution operator to *PDE*s as well as in computer vision for classification, in-painting, and segmentation (Li et al., 2020a; Kovachki et al., 2021a; Guibas et al., 2021). By leverage spectral theory, *FNO*s have successfully advanced frontiers in weather forecasts, carbon storage, and seismology (Pathak et al., 2022; Wen et al., 2022; Yang et al., 2021). While *FNO*s have shown tremendous speed-up over classical numerical methods, their efficacy can be limited due to the rapid growth in memory needed to represent complex operators. This growth may become a bottleneck in their application to high-resolution physical simulations such as climate or materials modeling. In general, despite significant speed up and better flexibility, prior works on neural operators suffer from similar memory complexity issues as conventional solvers do on high-resolution problems.

In the worst case, the large memory complexity is required and, in fact, is unavoidable due to the need for resolving fine scale features globally. However, many real-world problems, possess local structure that is not currently exploited by neural operator methods.For instance, consider a weather forecast where predictions for the next hour are heavily dependent on the weather conditions in local regions and minimally on global weather conditions. Incorporating and learning this local structure of the underlying *PDE*s is the keys to overcoming the curse of memory complexity.

In this work, we propose a new, scalable neural operator that addresses these issues by leveraging the structure in both the domain space and the parameter space, Figure 1. Specifically, we introduce the multi-grid tensor operator (*MG-TFNO*), a model that exploits locality in physical space by a novel multi-grid domain decomposition approach to compress the input domain size by up to $7\times$ while leveraging the global interactions of the model parameters to compress them by up to $200\times$ without any loss of accuracy (see Table 1).

**In the input space**, to predict the solution in any region of the domain, *MG-TFNO* decomposes the input domain into small local regions to which hierarchical levels of global information are added in a multi-grid fashion. Since a local prediction depends most strongly on its immediate spatial surroundings, the farther field information is downsampled to lower resolutions, progressively, based on its distance from the region of interest. Thus, *MG-TFNO* achieves a low memory complexity by using high resolution data only locally and coarse resolution data globally. Due to its state-of-the-art performance on *PDE* problems and efficient FFT-based implementation, we use the *FNO* as the backbone architecture for our method.

**In the parameter space**, we exploit the spatio-temporal structure of the underlying *PDE* solution operator by parameterizing the convolutional weights within the Fourier domain with a low-rank tensor factorization. Specifically, we impose a coupling between all the weights in the Fourier space by jointly parameterizing them with a single tensor, learned in a factorized form such as Tucker or Canonical-Polyadic (Kolda & Bader, 2009). This coupling allows us to limit the number of parameters in the model without limiting its expressivity. On the contrary, this low-rank regularization on the model mitigates over-fitting and improves generalization. Intuitively, our method can be thought of as a fully-learned implicit scheme capable of converging in a very few fixed number of iterations. Due to the global nature of the integral kernel transform, the *FNO* avoids the Courant–Friedrichs–Lewy (CFL) condition plaguing explicit schemes, allowing convergence in only a few steps (Courant et al., 1928). Our weight coupling ensures maximum communication between the steps, mitigating possible redundancies in the learned kernels and reducing the complexity of the optimization landscape.

**In summary, we make the following contributions:**

- **We propose** *MG-TFNO*, a novel neural operator parameterized in the spectral domain by a single low-rank factorized tensor, allowing its size to grow linearly with the size of the problem.

- **Our tensor operator achieve better performance with a fraction of the parameters**: we outperform *FNO* on solving the turbulent Navier Stokes equations with more than $400\times$ weight compression ratio, Figure 4a.

- **Our method overfits less and does better in the low-data regime**. In particular, it outperforms *FNO* with less than half the training samples, Figure 5.

- **We introduce a novel multi-grid domain decomposition approach**, a technique which allows the operator to predict the output only on local portions of the domain thus reducing the memory usage by an order of magnitude with no performance degradation.

- **Combining tensorization with multi-grid domain decomposition leads to our best model**, *MG-TFNO*, which is more efficient in terms of task performance, computation, and memory. *MG-TFNO* achieves $2.5\times$ lower error with $10\times$ model weight compression, and $1.8\times$ domain compression, Table 1.

## 2 BACKGROUND

Here, we review related works and introduce the background necessary to explain our approach.

**Neural Operators.** Many physical phenomena are governed by *PDE*s and a wide range of scientific and engineering computation problems are based on solving these equations. In recent years, a new perspective to *PDE*s dictates to formulate these problems as machine learning problems where solutions to *PDE*s are learned. Prior works mainly focused on using neural networks to train for the solution map of *PDE*s (Guo et al., 2016; Zhu & Zabaras, 2018; Adler & Oktem, 2017; Bhatnagar et al., 2019). The use of neural networks in the prior works limits them to a fixed grid and narrows their applicability to *PDE*s where maps between function spaces are desirable. Multiple attempts

have been made to address this limitation. For example mesh free methods are proposed that locally output mesh-free solution (Esmaeilzadeh et al., 2020), but they are local and still limited to fixed input gird.

A new deep learning paradigm, neural operators, are proposed as maps between function spaces (Li et al., 2020b; Kovachki et al., 2021b). They are discretization invariants maps. The input functions to neural operators can be presented in any discretization, mesh, resolution, or basis. The output functions can be evaluated at any point in the domain. Variants of neural operators deploy a variety of Nyström approximation to develop new neural operator architecture. Among these, multi-pole neural operators (Li et al., 2020c) utilize the multi-pole approach to develop computationally efficient neural operator architecture. Inspired by the spectral method, Fourier-based neural operators show significant applicability in practical applications (Li et al., 2020a; Yang et al., 2021; Wen et al., 2022; Rahman et al., 2022a). Principle component analysis, wavelet bases, and u-shaped methods are also considered (Gupta et al., 2021; Bhattacharya et al., 2020; Liu et al., 2022; Rahman et al., 2022b; Yang et al., 2022). It is also shown that neural operators can solely be trained using *PDE*s, resulting in physics-informed neural operators, opening new venues for hybrid data and equation methods (Li et al., 2021b) to tackle problems in scientific computing.

**Tensor methods in deep learning.** As we move beyond learning from simple structures to solving increasingly complex problems, the data we manipulate becomes more structured. To efficiently manipulate these structures, we need to go beyond matrix algebra and leverage the spatio-temporal stucture. For all purposes of this paper, tensors are multi-dimensional arrays and generalize the concept of matrices to more than 2 modes (dimensions). For instance, RGB images are encoded as third-order (three dimensional) tensors, videos are $4^{\text{th}}$ order tensors and so on and so forth. Tensor methods methods generalize linear algebraic methods to these higher-order structures. They have been very successful in various application in computer vision, signal processing, data mining and machine learning (Panagakis et al., 2021; Janzamin et al., 2019; Sidiropoulos et al., 2017; Papalexakis et al., 2016).

Using tensor decomposition Kolda & Bader (2009), previous works have been able to compress and improve deep networks for vision tasks. Either a weight matrix is tensorized and factorized Novikov et al. (2015), or tensor decomposition is directly to the convolutional kernels before fine-tuning to recover-for lost accuracy, which also allows for an efficient reparametrization of the network (Lebedev et al., 2015; Kim et al., 2016; Gusak et al., 2019). There is a tight link between efficient convolutional blocks and tensor factorization and factorized higher-order structures (Kossaifi et al., 2020). Similar strategies have been applied to multi-task learning (Bulat et al., 2020) and NLP (Papadopoulos et al., 2022; Cordonnier et al., 2020). Of all these prior works, none has been applied to neural operator. In this work, we propose the first application of tensor compression to learning operators and propose a Tensor OPerator (*TFNO*).

## 3 METHODOLOGY

Here, we briefly review operator learning as well as the Fourier Neural Operator, on which we build to introduce our proposed Tensor OPerator (*TFNO*) as well as the Multi-Grid Domain Decomposition, which together form our proposed *MG-TFNO*.

### 3.1 OPERATOR LEARNING

Let $\mathcal{A} := \{a : D_{\mathcal{A}} \to \mathbb{R}^{d_{\mathcal{A}}}\}$ and $\mathcal{U} := \{u : D_{\mathcal{U}} \to \mathbb{R}^{d_{\mathcal{U}}}\}$ denote two input and output function spaces respectively. Each function $a$, in the input function space $\mathcal{A}$, is a map from a bounded, open set $D_{\mathcal{A}} \subset \mathbb{R}^d$ to the $d_{\mathcal{A}}$-dimensional Euclidean space. Any function in the output function space $\mathcal{U}$ is a map from a bounded open set $D_{\mathcal{U}} \subset \mathbb{R}^d$ to the $d_{\mathcal{U}}$-dimensional Euclidean space. In this work we consider the case $D = D_{\mathcal{A}} = D_{\mathcal{U}} \subset \mathbb{R}^d$.

We aim to learn an operator $\mathcal{G} : \mathcal{A} \to \mathcal{U}$ which is a mapping between the two function spaces. In particular, given a dataset of $N$ points $\{(a_j, u_j)\}_{j=1}^N$, where the pair $(a_j, u_j)$ are functions satisfying $\mathcal{G}(a_j) = u_j$, we build an approximation of the operator $\mathcal{G}$. As a backbone operator learning model, we use neural operators as they are consistent and universal learners in function spaces. For

an overview of theory and implementation, we refer the reader to Kovachki et al. (2021b). We specifically use the *FNO* and give details in the forthcoming section (Li et al., 2020a).

## 3.2 FOURIER NEURAL OPERATORS

For simplicity, we will work on the $d$-dimensional unit torus $\mathbb{T}^d$ and first describe a single, pre-activation *FNO* layer mapping $\mathbb{R}^m$-valued functions to $\mathbb{R}^n$-valued functions. Such a layer constitutes the mapping $\mathcal{G} : L^2(\mathbb{T}^d; \mathbb{R}^m) \to L^2(\mathbb{T}^d; \mathbb{R}^n)$ defined as

$$\mathcal{G}(v) = \mathcal{F}^{-1}\big(\mathcal{F}(\kappa) \cdot \mathcal{F}(v)\big), \qquad \forall\, v \in L^2(\mathbb{T}^d; \mathbb{R}^m) \tag{1}$$

where $\kappa \in L^2(\mathbb{T}^d; \mathbb{R}^{n \times m})$ is a function constituting the layer parameters and $\mathcal{F}, \mathcal{F}^{-1}$ are the Fourier transform and its inverse respectively. The Fourier transform of the function $\kappa$ is parameterized directly by some fixed number of Fourier nodes denoted $\alpha \in \mathbb{N}$.

To implement (1), $\mathcal{F}, \mathcal{F}^{-1}$ are replaced by the discrete fast Fourier transforms $\hat{\mathcal{F}}, \hat{\mathcal{F}}^{-1}$. Let $\hat{v} \in \mathbb{R}^{s_1 \times \cdots \times s_d \times m}$ denote the evaluation of the function $v$ on a uniform grid discretizing $\mathbb{T}^d$ with $s_j \in \mathbb{N}$ points in each direction. We replace $\mathcal{F}(\kappa)$ with a weight tensor $\mathbf{T} \in \mathbb{C}^{s_1 \times \cdots \times s_d \times n \times m}$ consisting of the Fourier modes of $\kappa$ which are parameters to be learned. To ensure that $\kappa$ is parameterized as a $\mathbb{R}^{n \times m}$-valued function with a fixed, maximum amount of wavenumbers $\alpha < \frac{1}{2} \min\{s_1, \cdots, s_d\}$ that is independent of the discretization of $\mathbb{T}^d$, we leave as learnable parameters only the first $\alpha$ entries of $\mathbf{T}$ in each direction and enforce that $\mathbf{T}$ have conjugate symmetry. In particular, we parameterize half the corners of the $d$-dimensional hyperrectangle with $2^{d-1}$ hypercubes with length size $\alpha$. That is, $\mathbf{T}$ is made up of the free-parameter tensors $\tilde{\mathbf{T}}_1, \cdots, \tilde{\mathbf{T}}_{2^{d-1}} \in \mathbb{C}^{\alpha \times \cdots \times \alpha \times n \times m}$ situated in half of the corners of $\mathbf{T}$. Each corner diagonally opposite of a tensor $\tilde{\mathbf{T}}_j$ is assigned the conjugate transpose values of $\tilde{\mathbf{T}}_j$. All other values of $\mathbf{T}$ are set to zero. This is illustrated in the middle-top part of Figure 1 for the case $d = 2$ with $\tilde{\mathbf{T}}_1$ and $\tilde{\mathbf{T}}_2$. We will use the notation $\mathbf{T}(k, \cdots) = \tilde{\mathbf{T}}_k$ for any $k \in [2^{d-1}]$. The discrete version of (1) then becomes the mapping $\hat{\mathcal{G}} : \mathbb{R}^{s_1 \times \cdots \times s_d \times m} \to \mathbb{R}^{s_1 \times \cdots \times s_d \times n}$ defined as

$$\hat{\mathcal{G}}(\hat{v}) = \hat{\mathcal{F}}^{-1}\big(\mathbf{T} \cdot \hat{\mathcal{F}}(\hat{v})\big), \qquad \forall\, \hat{v} \in \mathbb{R}^{s_1 \times \cdots \times s_d \times m} \tag{2}$$

where the $\cdot$ operation is simply the matrix multiplication contraction along the last dimension. Specifically, we have

$$\big(\mathbf{T} \cdot \hat{\mathcal{F}}(\hat{v})\big)(l_1, \ldots, l_d, j) = \sum_{i=1}^{m} \mathbf{T}(l_1, \ldots, l_d, j, i)\big(\hat{\mathcal{F}}(\hat{v})\big)(l_1, \ldots, l_d, i) \tag{3}$$

From (2), a full FNO layer is build by adding a point-wise linear action to $\hat{v}$, a bias term, and applying a non-linear activation. In particular, from an input $\hat{v} \in \mathbb{R}^{s_1 \times \cdots \times s_d \times m}$, the output $\hat{q} \in \mathbb{R}^{s_1 \times \cdots \times s_d \times n}$ is given as

$$\hat{q}(l_1, \cdots, l_d, :) = \sigma\big(\mathbf{Q}\hat{v}(l_1, \cdots, l_d, :) + \hat{\mathcal{G}}(\hat{v}) + b\big)$$

with $\sigma : \mathbb{R} \to \mathbb{R}$ a fixed, non-linear activation, and $b \in \mathbb{R}^n$, $\mathbf{Q} \in \mathbb{R}^{n \times m}$, $\tilde{\mathbf{T}}_1, \cdots, \tilde{\mathbf{T}}_{2^{d-1}} \in \mathbb{C}^{\alpha \times \cdots \times \alpha \times n \times m}$ are the learnable parameters of the layer. The full FNO model consists of $L \in \mathbb{N}$ such layers each with weight tensors $\mathbf{T}_1, \cdots, \mathbf{T}_L$ that have learnable parameters $\tilde{\mathbf{T}}_k^{(l)} = \mathbf{T}_l(k, \cdots)$ for any $l \in [L]$ and $k \in [2^{d-1}]$. In the case $n = m$ for all layers, we introduce the joint parameter tensor $\mathbf{W} \in \mathbb{C}^{\alpha \times \cdots \times \alpha \times n \times n \times 2^{d-1}L}$ so that

$$\mathbf{W}\big(\ldots, 2^{d-1}(l-1) + k + 1\big) = \tilde{\mathbf{T}}_k^{(l)}.$$

Perusal of the above discussion reveals that there are $(2^d \alpha^d + 1)mn + n$ total parameters in each FNO layer. Note that, since $m$ and $n$ constitute the respective input and output channels of the layer, the number of parameters can quickly explode due the exponential scaling factor $2^d \alpha^d$ if many wavenumbers are kept. Preserving a large number of modes could be crucial for applications where the spectral decay of the input or output functions is slow such as in image processing or the modeling of multi-scale physics. In the following section we describe a tensorization method that is able to mitigate this growth without sacrificing approximation power.

### 3.3 TENSOR OPERATORS

In the previous section, we introduced a unified formulation of FNO where the whole operator is parametrized by a single parameter tensor $\mathbf{W}$. This enables us to introduce the tensor operator, which parameterizes efficiently $\mathbf{W}$ with a low-rank, tensor factorization. In this paper, we focus mostly on the Tucker decomposition, for its flexibility, but other decompositions such as Canonical Polyadic can be readily integrated. This joint parametrization has several advantages: i) it applies a low-rank constraint on the entire tensor $\mathbf{W}$, thus regularizing the model. These advantages translate into i) a huge reduction in the number of parameters, ii) better generalization and an operator less prone to overfitting. We show superior performance for low-compression ratios (up to $200\times$) and very little performance degradation when largely compressing ($> 450\times$) the model, iii) better performance in low-data regime.

In practice, we express $\mathbf{W}$ in a low-rank factorized form, e.g. Tucker or CP. In the case of a Tucker factorization with rank $(R_1, \cdots, R_d, R_L, R_I, R_O)$, where $R_L$ controls the rank across layers, $R_I = R_O$ control the rank across the input and output co-dimension, respectively, and $R_1, \cdots, R_d$ control the rank across the dimensions of the operator:

$$\mathbf{W} = \sum_{r_1=1}^{R_1} \cdots \sum_{r_d=1}^{R_d} \sum_{r_i=1}^{R_I} \sum_{r_o=1}^{R_O} \sum_{r_l=1}^{R_L} \mathbf{G}(r_1, \cdots, r_d, r_i, r_o, r_l) \cdot \mathbf{U^{(1)}}(:,r_1) \cdot \ \cdots \ \cdot \mathbf{U^{(d)}}(:,r_d) \cdot$$
$$\mathbf{U^{(I)}}(:,r_i) \cdot \mathbf{U^{(O)}}(:,r_o) \cdot \mathbf{U^{(L)}}(:,r_l).$$

Here, $\mathbf{G}$ is the core of size $R_L \times R_I \times R_O \times R_1 \times \cdots \times R_d$ and $\mathbf{U^{(L)}}, \mathbf{U^{(I)}}, \mathbf{U^{(O)}}, \mathbf{U^{(1)}}, \cdots, \mathbf{U^{(d)}}$ are factor matrices of size $(R_L \times L), (R_I \times I), (R_O \times O), (R_1 \times \alpha), \cdots, (R_d \times \alpha)$, respectively.

Figure 2: **Illustration of a Tucker decomposition.** For clarity , we show $\mathbf{W}$ as a $3^{\text{rd}}$-order tensor weight.

Note that the mode (dimension) corresponding to the co-dimension can be left uncompressed, by setting $R_L = L$ and $\mathbf{U^{(L)}} = \text{Id}$. This leads to a layerwise compression. Also note that having a rank of 1 along any of the modes would mean that the slices along that mode differ only by a (multiplicative) scaling parameter. Also note that during the forward pass, we can pass $\mathbf{T}$ directly in factorized form to each layer by selecting the corresponding rows in $\mathbf{U^{(L)}}$. The contraction in equation 3 is then either done using the reconstructed tensor, or by directly contracting $\hat{\mathcal{F}}(\hat{v})$ with the factors of the decomposition.

This joint factorization along the entire operator allows us to leverage redundancies both locally and across the entire operator. This leads to much reduced memory footprint, with only a fraction of the parameter. It also acts as a low-rank regularizer on the operator, facilitating training. Finally, through the global parametrization, we introduce skip connections that allow gradient to flow through the latent parametrization to all the layers jointly, leading to better optimization.

This formulation is general and works with any tensor factorization. For instance, we also explore a Canonical-Polyadic decomposition (CP) which can be seen as a special case of Tucker with a super-diagonal core. In that case, we set a single rank $R$ and express the weights as a weighted sum of $R$ rank-1 tensors. Concretely:

$$\mathbf{W} = \sum_{r=1}^{R} \lambda_r \mathbf{U^{(1)}}(:,r) \cdot \ \cdots \ \cdot \mathbf{U^{(d)}}(:,r) \cdot \mathbf{U^{(I)}}(:,r) \cdot \mathbf{U^{(O)}}(:,r) \cdot \mathbf{U^{(L)}}(:,r). \quad (4)$$

where $\mathbf{U^{(L)}}, \mathbf{U^{(I)}}, \mathbf{U^{(O)}}, \mathbf{U^{(1)}}, \cdots, \mathbf{U^{(d)}}$ are factor matrices of size $(R \times L), (R \times I), (R \times O), (R \times \alpha), \cdots, (R \times \alpha)$, respectively and $\lambda \in \mathbb{R}^{\mathbf{R}}$. Note that the CP, contrarily to the Tucker, has a single rank parameter, shared between all the dimensions. This means that to maintain the number of parameters the same, $R$ needs to be very high, which leads to memory issues. This makes CP more suitable for large compression ratios, and indeed, we found it leads to better performance at high-compression / very low-rank.

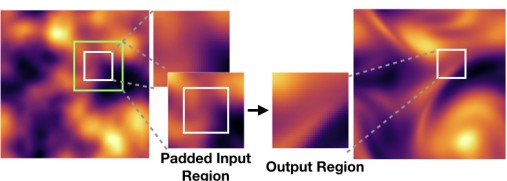
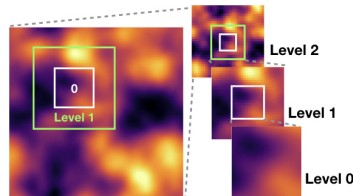

(a) **Predicting with padded regions.** Local region in the input is padded and used to predict the corresponding region in the output.

(b) **MG-Domain Decomposition.** Progressively larger spatial regions are added to a local region by subsampling.

Figure 3: **Domain decomposition in space (3a) and our Multi-Grid based approach. (3b)**. White squares represent the region of interest while yellow squares the larger embeddings.

### 3.4 MULTI-GRID DOMAIN DECOMPOSITION

Having introduced our decomposition in the operator's parameter space, we now introduce our novel multi-grid appraoch to decompose the problem domain.

**Domain decomposition** is a method commonly used to parallelize classical solvers for time-dependent PDEs that is based on the principal that the solution for a fixed local region in space depends mostly on the input at the same local region (Chan & Mathew, 1994). In particular, since the time-step $h > 0$ of the numerical integrator is small, the solution $u(x, t+h)$, for any point $x \in D$ and $t \in \mathbb{R}_+$, depends most strongly on the points $u(y, t)$ for all $y \in B(x, r(h))$ where $B(x, r(h))$ denotes the ball centered at $x$ with radius $r(h)$. This phenomenon is easily seen for the case of the heat equation where, in one dimension, the solution satisfies

$$u(x, t + h) \propto \int_{-\infty}^{\infty} \exp\left(\frac{-(x-y)^2}{4h}\right) u(y, t) \, \mathrm{d}y \approx \int_{x-4h}^{x+4h} \exp\left(\frac{-(x-y)^2}{4h}\right) u(y, t) \, \mathrm{d}y$$

with the approximation holding since 99.9937% of the kernel's mass is contained within $B(x, 4h)$. While some results exist, there is no general convergence theory for this approach, however its empirical success has made it popular for various numerical methods (Albin & Bruno, 2011).

To exploit this localization , the domain $D$ is split in $q \in \mathbb{N}$ pairwise-disjoint regions $D_1, \cdots, D_q$ so that $D = \cup_{j=1}^q D_j$. Each region $D_j$ is then embedded into a larger one $Z_j \supset D_j$ so that points away from the center of $D_j$ have enough information to be well approximated. A model can then be trained so that the approximation $\mathcal{G}(a|_{Z_j})|_{D_j} \approx u|_{D_j}$ holds for all $j \in [q]$. This idea is illustrated in Figure 3a where $D = [0, 1]^2$ and all $D_j, Z_j$ are differently sized squares. This allows the model to be ran fully in parallel hence its time and memory complexities are reduced linearly in $q$.

**Multi-Grid.** Domain decomposition works well in classical solvers when the time step $h > 0$ is small because the mapping $u(\cdot, t) \mapsto u(\cdot, t+h)$ is close to the identity. However the major advancement made by machine learning based operator methods for PDEs is that a model can approximate the solution, in one shot, for very large times i.e. $h > 1$. But, for larger $h$, the size of $Z_j$ relative to $D_j$ must increase to obtain the same approximation accuracy, independently of model capacity. This causes any computational savings made by the decomposition approach to be lost.

To mitigate this, we propose a multi-grid based domain decomposition approach where global information is added hierarchically at different resolutions. While our approach is inspired by the classical multi-grid method, it is not based on the V-cycle algorithm (McCormick, 1985). For ease of presentation, we describe this concept when a domain $D = \mathbb{T}^2$ is uniformly discretized by $2^s \times 2^s$ points, for some $s \in \mathbb{N}$, but note that generalizations can readily be made. Given a final level $L \in \mathbb{N}$, we first sub-divide the domain into $2^{2L}$ total regions each of size $2^{s-L} \times 2^{s-L}$ and denote them $D_1^{(0)}, \cdots, D_{2^{2L}}^{(0)}$. We call this the zeroth level. Then, around each $D_j^{(0)}$, for any $j \in [2^{2L}]$, we consider the square $D_j^{(1)}$ of size $2^{s-L+1} \times 2^{s-L+1}$ that is equidistant, in every direction, from each boundary of $D_j^{(0)}$. We then subsample the points in $D_j^{(1)}$ uniformly by a factor of $\frac{1}{2}$ in each direction, making $D_j^{(1)}$ have $2^{s-L} \times 2^{s-L}$ points. We call this the first level. We continue this process by considering the squares $D_j^{(2)}$ of size $2^{s-L+2} \times 2^{s-L+2}$ around each $D_j^{(1)}$ and subsample them

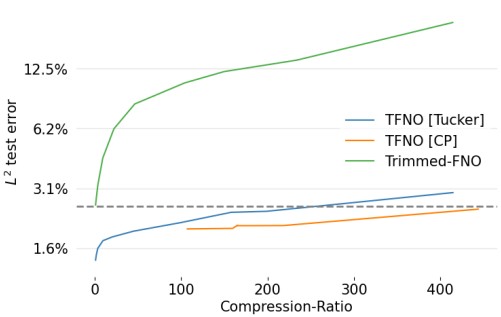

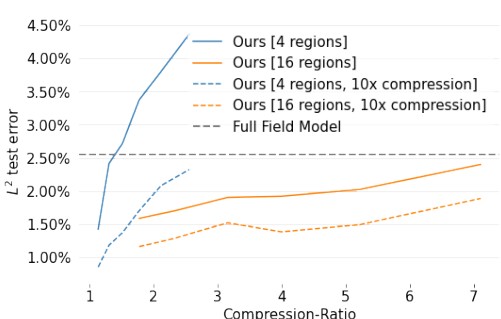

(a) **Tensorization: error in logscale as a function of the compression ratio.** We compare the tensor neural operator with an FNO with the same number of parameters (*trimmed*). **We achieve over 200x compression ratio with better performance that the original FNO with Tucker, and over 400x compression ratio with CP.**

(b) **MG-Domain Decomposition: error as a function of the domain compression ratio.** We compare *MG-TFNO* with different numbers of multigrid regions both with and without weight tensor compression to a full field FNO model. **We achieve over 7x input space compression and 10x parameter space compression ratios with better performance that the original FNO.**

Figure 4: **Impact of Tensorization (4a) and Multi-Grid Domain Decomposition (4b)**

uniformly by a factor of $\frac{1}{4}$ in each direction to again yield squares with $2^{s-L} \times 2^{s-L}$ points. The process is repeated until the $L$th level is reached wherein $D_j^{(L)}$ is the entire domain subsampled by a factor of $2^{-L}$ in each direction. The process is illustrated for the case $L = 2$ in Figure 3b. Since we work with the torus, the region of the previous level is always at the center of the current level.

The intuition behind this method is that since the dependence of points inside a local region diminishes the further we are from that region, it is enough to have coarser information, as we go farther. We combine this multi-grid method with the standard domain decomposition approach by building appropriately padded squares $Z_j^{(l)}$ of size $2^{s-L} + 2p \times 2^{s-L} + 2p$ around each $D_j^{(l)}$ where $p \in \mathbb{N}$ is the amount of padding to be added in each direction. We then take the evaluations of the input function $a$ at each level and concatenate them as channels. In particular, we train a model so that $\hat{\mathcal{G}}\big((a|_{Z_j^{(0)}}, \cdots, a|_{Z_j^{(L)}}))\big)|_{D_j^{(0)}} \approx u|_{D_j^{(0)}}$. Since the model only operates on each padded region separately, we reduce the total number of grid points used from $2^{2s}$ to $(2^{s-L} + 2p)^2$ and define the domain compression ratio as the quotient of these numbers. Furthermore, note that, assuming $a$ is $\mathbb{R}^{d_{\mathcal{A}}}$-valued, a model that does not employ our multi-grid domain decomposition uses inputs with $d_{\mathcal{A}}$ channels while our approach builds inputs with $(L + 1)d_{\mathcal{A}}$ channels. In particular, the number of input channels scale only logarithmically in the number of regions hence global information is added at very little additional cost. Indeed, FNO models are usually trained with internal widths much larger than $d_{\mathcal{A}}$ hence the extra input channels cause almost no additional memory overhead.

## 4 EXPERIMENTS

**Data.** We experiment on a dataset of 10K training samples and 2K test samples of the two-dimensional Navier-Stokes equation with Reynolds number 500. We also experiment with the one-dimensional viscous Burgers' equation. Details about the datasets as well the results on Burgers are given in Appendices A.2 and A.5 respectively.

**Training the operator.** Since *MG-TFNO* predicts local regions which are then stitched together to form a global function without any communication, aliasing effects can occur where one output prediction does not flow smoothly into the next. To prevent this, we train our model using the $H^1$ Sobolev norm (Czarnecki et al., 2017; Li et al., 2021a). By matching derivatives, training with this loss prevents any discontinuities from occurring and the output prediction is smooth.

**Tensorizing: better compression.** In Figure 4a, we show the performance of our approach (TNO) compared to the original *FNO*, for varying compression ratios. In the Trimmed-*FNO*, we adjust the

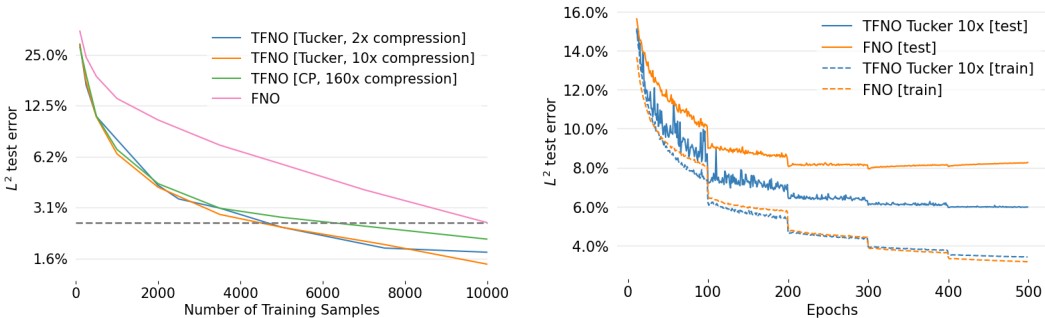

Figure 5: **Error as a function of the number of training samples (left) and training VS testing loss.** We compare *TFNO* with a regular *FNO*. Note that on the left we show the testing $L^2$ error while, for training, the $H^1$ loss is used and that is compared with the $H^1$ test-error on the right.

width in order to match the number of parameters in our TNO. Our method massively outperforms the Trimmed-*FNO* at every single fixed parameter amount. Furthermore, even for very large compression ratios, our FNO outperforms the full-parameter FNO model. This is likely due to the regularizing effect of the tensor factorization on the weight, showing that many of the ones in the original model are redundant.

**Tensorizing: better generalization.** Figure 5 (left) shows that our TNO generalizes better with less training samples. Indeed, at every fixed amount of training samples, the TNO massively outperforms the full-parameter FNO model. Even when only using half the samples, our TNO outperforms the FNO trained on the full dataset. Furthermore, Figure 5 (right) shows that our TNO overfits significantly less than FNO, demonstrating the regularizing effect of the tensor decomposition. This result is invaluable in the *PDE* setting where we frequently have very little training samples available due to the high computational cost of traditional *PDE* solvers.

**Multi-Grid Domain Decomposition.** We show the impact of multi-grid domain decomposition on performance in Figure 4b. We find that lower compression ratios (corresponding to a larger amount of padding in the decomposed regions) perform better which is unsurprising since more information is incorporated into the model. More surprisingly, we find that using a larger number of regions (16) performs consistently better than using a smaller number (4) and both can outperform the full-field FNO. This can be due to the fact that: i) the domain decomposition acts as a form of data augmentation, exploiting the transnational invariance of the *PDE* and more regions yield larger amounts of data, and ii) the output space of the model is simplified since a function can have high frequencies globally but may only have low frequencies locally. Consistently, we find that the tensor compression in the weights acts as regularizer and improves performance across the board.

**Putting it all together:** *MG-TFNO*. Tensorization and multi-grid domain decomposition not only improve performance individually, but their advantages compound and lead to a strictly better algorithm that scales well to higher-resolution data by decreasing the number of parameters in the model as well as the size of the inputs thereby improving performance as well as memory and computational footprint. Table 1 shows a comparison of FNO with Tensorization alone, multi-grid domain decomposition alone, and our joint approach combining the two, *MG-TFNO*. Our results imply that, under full-parallelization, the memory footprint of the model's inference can be reduced by $7\times$ and size of its weights by $10\times$ while also improving performance.

## 5 CONCLUSION

In this work, we introduced a novel tensor operator (*TFNO*) as well as a multi-grid domain decomposition approach which together form *MG-TFNO*, an operator model that outperforms the *FNO* with a fraction of the parameters and memory complexity requirements. *MG-TFNO* scales better, generalizes better, and requires fewer training samples to reach the same performance. In our future work, we plan to deploy *MG-TFNO* and tackle very high-resolution large-scale weather forecasts for which existing deep learning models are prohibitive.

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

## A  APPENDIX

### A.1  NOTATION

We summarize the notation used throughout the paper in Table 2.

| Variable | Meaning | Dimensionality |
|:---:|:---:|:---:|
| $\mathbf{T}$ | Tensor of weights in the Fourier domain | $\mathbb{C}^{\alpha \times \cdots \times \alpha \times m \times n}$ |
| $\mathbf{W}$ | Weight tensor parameterizing the entire operator | $\mathbb{C}^{\alpha \times \cdots \times \alpha \times n \times n \times 2^{d-1}L}$ |
| $\mathcal{A}$ | Input function space | Infinite |
| $\mathcal{U}$ | output function space | Infinite |
| $a$ | Input function | Infinite |
| $u$ | Output function | Infinite |
| $D_{\mathcal{A}}$ | Domain of function a | $d$ |
| $D_{\mathcal{U}}$ | Domain of function u | $d$ |
| $d_{\mathcal{A}}$ | Dimension of the co-domain of the input functions | 1 |
| $d_{\mathcal{U}}$ | Dimension of the co-domain of the output functions | 1 |
| $\mathcal{F}$ | Fourier transform | Infinite |
| $\mathcal{F}^{-1}$ | Fourier transform | Infinite |
| $L$ | Number of integral operation layers | In $\mathbb{N}$ |
| $l$ | Layer index | Between 1 and $L$ |
| $\sigma$ | Point-wise activation operation | Infinite |
| $b$ | Bias vector | |
| $v$ | Function at each layer | Infinite |
| $\alpha$ | Number of kept frequencies in Fourier space | Between 1 and $\frac{1}{2}\min\{s_1, \cdots, s_d\}$ |

Table 2: **Table of notation**

## A.2 DATA

In this section, we introduce in detail the datasets used for our experiments.

**Navier-Stokes.** We consider the vorticity form of the two-dimensional Navier-Stokes equation,

$$
\begin{aligned}
\partial_t \omega + \nabla^{\perp}\phi \cdot \omega &= \frac{1}{\text{Re}}\Delta\omega + f, \quad x \in \mathbb{T}^2, \, t \in (0, T] \\
-\Delta\phi &= \omega, \quad \int_{\mathbb{T}^2}\phi = 0, \quad x \in \mathbb{T}^2, \, t \in (0, T]
\end{aligned}
\tag{5}
$$

with initial condition $\omega(0, \cdot) = 0$ where $\mathbb{T}^2 \cong [0, 2\pi]^2$ is the torus, $f \in \dot{L}^2(\mathbb{T}^2; \mathbb{R})$ is a forcing function, and Re $> 0$ is the Reynolds number. Then $\omega(t, \cdot) \in \dot{H}^s(\mathbb{T}^2; \mathbb{R})$ for any $t \in (0, T]$ and $s > 0$, is the unique weak solution to (5) (Temam, 1988). We consider the non-linear operator mapping $f \mapsto \omega(T, \cdot)$ with $T = 5$ and fix the Reynolds number Re $= 500$. We define the Gaussian measure $\mu = \mathcal{N}(0, C)$ on the forcing functions where we take the covariance $C = 27(-\Delta + 9I)^{-4}$, following the setting in (De Hoop et al., 2022). Input data is obtained by generating i.i.d. samples from $\mu$ by a KL-expansion onto the eigenfunctions of $C$ (Powell et al., 2014). Solutions to (5) are then obtained by a pseudo-spectral scheme (Chandler & Kerswell, 2013).

**Burgers' Equation.** We consider the one-dimensional Burgers' equation on the torus,

$$
\begin{aligned}
\partial_t u + u u_x &= \nu u_{xx}, \quad x \in \mathbb{T}, \, t \in (0, T] \\
u|_{t=0} &= u_0, \quad x \in \mathbb{T}
\end{aligned}
\tag{6}
$$

for initial condition $u_0 \in L^2(\mathbb{T}; \mathbb{R})$ and viscosity $\nu > 0$. Then $u(t, \cdot) \in H^s(\mathbb{T}; \mathbb{R})$, for any $t \in \mathbb{R}_+$ and $s > 0$, is the unique weak solution to 6 (Evans, 2010). We consider the non-linear operator $u_0 \mapsto u(T, \cdot)$ with $T = 0.5$ or 1 and fix $\nu = 0.01$. We define the Gaussian measure $\mu = \mathcal{N}(0, C)$ where we take the covariance $C = 3^{5/2}(-\frac{d^2}{dx^2} + 9I)^{-3}$. Input data is obtained by generating i.i.d. samples from $\mu$ by a KL-expansion onto the eigenfunctions of $C$. Solutions to (6) are then obtained by a pseudo-spectral solver using Heun's method. We use 8K samples for training and 2K for testing.

## A.3 TENSOR OPERATOR: TENSOR DECOMPOSITION

In this section, we expand further on the tensor decomposition composing the tensor operator.

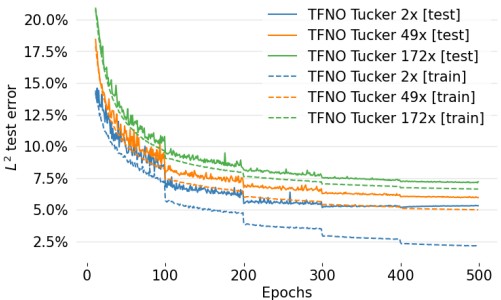
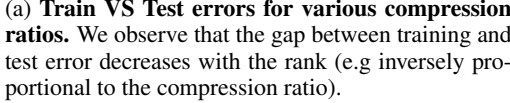
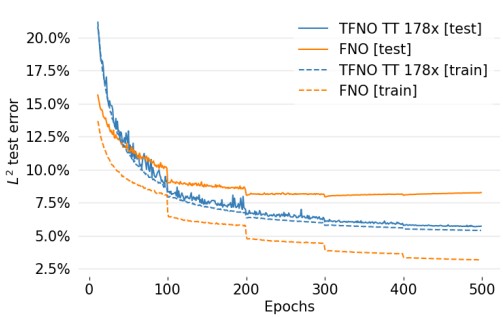

(a) **Train VS Test errors for various compression ratios.** We observe that the gap between training and test error decreases with the rank (e.g inversely proportional to the compression ratio).

(b) **Train VS Test error over time for a TOP with a TT factorization.**

Figure 6: **Impact of the rank for a TOP Tucker (6a) and train/test curve for a TOP-TT (6b)**

We first rewrite the entire weight parameter for the Tucker case, equivalently, using the more compact n-mode product as:

$$\mathbf{W} = \mathbf{G} \times_1 \mathbf{U^{(1)}} \cdots \times_d \mathbf{U^{(d)}} \times_{d+1} \mathbf{U^{(I)}} \times_{d+2} \mathbf{U^{(O)}} \times_{d+3} \mathbf{U^{(L)}}$$

**iFFT of the tensorized kernel** For any layer $l$, the $(j_1, j_2)$ coordinate of the matrix-valued convolution function $\kappa(x)$ is as follows,

$$
\begin{aligned}
[\kappa_l(x)]j1, j_2 = \sum_{i_1=1}^{m_1} \cdots \sum_{i_d=1}^{m_d} \sum_{r_l=1}^{R_L} \sum_{r_i=1}^{R_I} \sum_{r_o=1}^{R_O} \sum_{r_1=1}^{R_1} \cdots \sum_{r_d=1}^{R_d} \mathbf{G}(r_1, \cdots, r_d, r_i, r_o, r_l) \cdot \\
\mathbf{U^{(1)}}(i_1, r_1) \cdots \mathbf{U^{(d)}}(i_d, r_d) \cdot \mathbf{U^{(I)}}(j_1, r_i) \cdot \cdot \mathbf{U^{(O)}}(j_2, r_o) \cdot \mathbf{U^{(L)}}(l, r_l) \cdot \\
\exp(2\pi \sum_{k=1}^{d} i x_k i_k)
\end{aligned}
$$

We also note that other tensor decomposition can be straightforwardly used in our framework, such as the tensor-train decomposition Oseledets (2011). A rank-$(1, R_1, \cdots, R_N, R_I, R_O, R_L, 1)$ TT factorization expresses $\mathbf{W}$ as:

$$\mathbf{W}(i_1, \cdots, i_d, i_c, i_o, i_l) = \mathbf{G}_1(i_1) \cdot \times \mathbf{G}_N(i_d) \mathbf{G}_I(i_c) \times \cdots \mathbf{G}_O(i_o) \times \cdots \mathbf{G}_L(i_l).$$

Where each of the factors of the decompositions $\mathbf{G}_k$ are third order tensors of size $R_k \times I_k \times R_{k+1}$. We show examples of *TFNO* trained with a TT factorization in the coming sections.

### A.4 IMPLEMENTATION DETAILS

**Implementation** We use PyTorch Paszke et al. (2017) for implementing all the models. The tensor operations are implemented using TensorLy and TensorLy-Torch Kossaifi et al. (2019); Kossaifi (2021). We will release the code and data used upon acceptance of the paper.

**Hyper-parameters** We train all models via gradient backpropagation using a mini-batch size of 16, the Adam optimizer, with a learning rate of $1e^{-3}$, weight decay of $1e^{-4}$, for 500 epochs, decreasing the learning rate every 100 epochs by a factors of $\frac{1}{2}$. The model width is set in all cases to 64 except when specified otherwise (for the Trimmed *FNO*), meaning that the input was first lifted (with a linear layer) from the number of input channels to that width. The projection layer projects from the width to 256 and a prediction linear layer outputs the predictions. 10000 samples were used for training, as well as a separate set of 2000 samples for test. For $\alpha$, we keep 40 Fourier coefficients for height and 24 for the width. All experiments are done on a NVIDIA Tesla V100 GPU.

| Method | $L^2$ test error | # Params | Model CR |
|---|---|---|---|
| **FNO** | 2.54% | 58 M | 0× |
| **TOP** [Tucker] | 2.36% | 366 K | 172× |
| **TOP** [CP] | 2.03% | 350 K | 179× |
| **TOP** [TT] | 1.83% | 353 K | 178× |

Table 3: **Relative $L^2$ test error of our *MG-TFNO* approach for different tensor decompositions** for a comparable compression ratio.

## A.5 ABLATION STUDIES

In this section, we further study the properties of our model through ablation studies. We first look at how *TFNO* suffers less from overfitting thank to the low-rank constraints, before comparing its performance with various tensor decompositions. Finally, we perform ablation studies for our multi-grid domain decomposition on Burger's equation.

### A.5.1 OVERFITTING AND LOW-RANK CONSTRAINT

Here, we show that lower-ranks (higher compressions), lead to reduced overfitting. In Figure 6a, we show the training and testing $H^1$ errors for our TOP with Tucker decompositions, at varying compression ratios (2x, 49x and 172x). We can see how, while the test error does not vary much, the gap between training and test errors reduces as we decrease the rank. As we can see, Tucker, while being the most flexible, does not perform as well at higher compression ratios, there CP and Tensor-Train lead to lower error.

### A.5.2 TENSOR-TRAIN AND TOP

Our approach is independent of the choice of tensor decomposition. We already showed how Tucker is most flexible and works well across all ranks. We also showed that CP, while memory demanding for high rank, leads to better performance and low-rank. Our method can also be used in conjunction with other decompositions such as tensor-train. To illustrate this, we show here the convergence behaviour of TNO with a Tensor-Train decomposition, for a compression ratio of 178, figure 6b.

We also compare in Table 3 our *TFNO* with different tensor decompositions.

### A.5.3 BURGERS' EQUATION

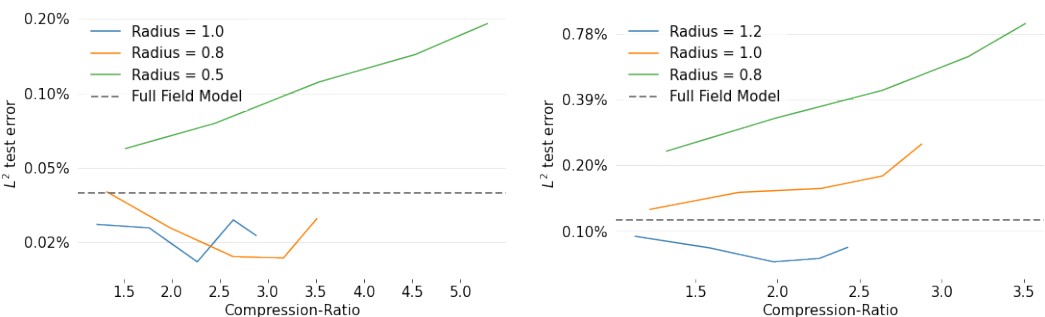

Figure 7: **Error on Burgers' equation with $T = 0.5$ (left) and $T = 1$ (right) as a function of domain compression ratio using standard domain decomposition without our multi-grid approach.** We evaluate the performance of the standard domain decomposition approach. The radius indicates the size, in physical space, of the padding added to each region.

We test the efficacy of the standard domain decomposition approach by training on two separate Burgers problems: one with a final time $T = 0.5$ and one with $T = 1$. As described in Section 3.4, we expect that for $T = 1$, each region requires more global information thus significantly more padding need to be used in order to reach the same error. The results of Figure 7 indeed confirms this.

The domain compression ratios needed for the approach to reach the performance of the full-field model are higher, indicating the need for incorporating global information. These results motivate our multi-grid domain decomposition approach.

