# OpenReview forum: "Multi-Grid Tensorized Fourier Neural  Operator for High Resolution PDEs"
_ICLR.cc/2023/Conference — Submitted to ICLR 2023_

### Official Review · Reviewer_zgdt · 2022-10-19

**Confidence:** 3
**Clarity, Quality, Novelty And Reproducibility:** The proposed method involves the Four…
**Correctness:** 3
**Technical Novelty And Significance:** 3
**Empirical Novelty And Significance:** 3
**Recommendation:** 5

**Strength And Weaknesses:**

Strength

MG-TFNO outperforms the FNO with a fraction of the parameters and memory complexity requirements.

Figure 1 clearly illustrates the structure of the network.

Weaknesses

How to handle the Fourier transform of non-periodic problems?

Will the computational cost of the Fourier transform be large?

How is the differentiation of the Fourier transform implemented?

Can this method make a 3D prediction?


**Summary Of The Paper:**

This paper proposes to predict solutions of PDE locally and unite them into a global solution. The proposed neural operator scales to large resolutions by leveraging local and global structures by decomposing both the input domain and the operator's parameter space. The proposed method can be trained with much fewer samples than previous approaches, outperforming the FNO when trained with just half the samples.

**Summary Of The Review:**

The paper proposes a new approach but solves only a few simple problems. There are already many similar methods, and it is difficult to say which is better and which is more practical. Studying large-scale examples in three dimensions may be more critical.

---

> ### Author Response · Authors · 2022-11-20
> **Response to Reviewer zgdt**
>
> We would like to thank reviewer zgdt for the thoughtful review.
>
> > *How to handle the Fourier transform of non-periodic problems?*
>
> The definition of Fourier transform does not require periodic functions as input. However, it is imperfectly motivated to pad the domain with zero functions to improve the performance, a similar practice common in celebrated convolutional neural networks. This practice has been first in the operator learning setting by (Li et al., 2020c). Furthermore, Fourier continuation type methods can be used to obtain smooth extension of the function which facilitate numerical differentiation with FFTs in non-periodic settings.
>
> > *Will the computational cost of the Fourier transform be large?*
>
> Since the data is provided on a discretization, the Fourier transform is approximated using the discrete Fourier transform, which can be computed using the fast Fourier transform (FFT) approach. While the computation complexity of general convolution is the order of O(n^2) for n the grid size, the computation complexity of FFT is only O(n log n) which scales favorably even for large n. In relation to standard convolutions implemented in deep learning libraries (which are not resolution invariant), it is more efficient to use FFTs when support of the kernel is larger than log n.
>
> > *How is the differentiation of the Fourier transform implemented?*
>
> We directly use the FFT implementation of PyTorch which computes the differentiation for the real and imaginary parts of complex numbers separately. The PyTorch functions use the efficient cuFFT implementation in the backend.
>
> > *Can this method make a 3D prediction?*
>
> Yes, the multi-grid idea proposed in this paper is independent of the domain dimension. Moreover, the tensorization approach taken in this paper shines even further when dealing with very high-dimensional domains. We will add these remarks to the paper.

---

> > ### Comment · Reviewer_zgdt · 2022-11-23
> > **Thank you for your reply**
> >
> > Thank you for your reply. Though the author has made efforts to address my concerns, the paper still doesn’t show its unique advantages over other methods. Unfortunately, I have to leave the score as it is.

---

> > > ### Author Response · Authors · 2022-11-29
> > > **Response to reviewer zgdt**
> > >
> > > > *Thank you for your reply. Though the author has made efforts to address my concerns, the paper still doesn’t show its unique advantages over other methods. Unfortunately, I have to leave the score as it is.*
> > >
> > > We thank the reviewer for responding to our rebuttal and try here to comment as best as possible on the following summary from the reviewer:
> > >
> > > > *The paper proposes a new approach but solves only a few simple problems. There are already many similar methods, and it is difficult to say which is better and which is more practical. Studying large-scale examples in three dimensions may be more critical.*
> > >
> > > Thank you for the prompt response; we appreciate the feedback and interest in our work.
> > >
> > > > *The paper proposes a new approach but solves only a few simple problems*
> > >
> > > We are happy to see that the reviewer acknowledges the novelty of our method. While we agree that many more problems can be tackled, we believe that the Navier-Stokes problem presented is not simple and constitutes a strong benchmark for various methods in the field. Indeed similar versions of it appear in [1,2], along with the references in our paper. While we hope that our method enables the use of machine learning based approaches for difficult, high-resolution engineering problems, the goal of our current work is to demonstrate its efficacy on a standard benchmark for which comparisons exist.
> > >
> > > > *There are already many similar methods, and it is difficult to say which is better and which is more practical. Studying large-scale examples in three dimensions may be more critical.*
> > >
> > > While we agree that many frameworks for solving PDEs with machine learning exist, the current consensus in the literature points to FNO being best in terms of accuracy and scaling. In particular, [1] includes a thorough comparison of FNO against other operator based methods on essentially the same data and shows FNO performs best in terms of accuracy while keeping favorable scaling. These findings are confirmed in [2]. In summary, FNO is the state-of-the-art for neural PDE solvers and we compare it to a highly-optimized version of FNO where we show large improvements. In addition, we enable scaling to resolutions not previously possible. While 3D is outside the scope of this paper, we do further include large-scale experiments. Firstly, all our experiments are on a resolution of 128x128 which is higher than most previous literature (usually most consider 64x64). Second, we include additional experiments on 512x512; please see response to reviewer J6V8. We copy them here for convenience:
> > >
> > > $\\begin{array} {|r|r|r|r|r|}\\hline
> > > Model & Width & Patches & Padding & Relative L^2 Error  \\\\ \\hline
> > > FNO & 12 & 0 & 0 & 6.1 \\% \\\\ \\hline
> > > MG-FNO & 42 & 4 & 70 & 2.9 \\% \\\\ \\hline
> > > MG-FNO &66 & 4 & 53 & 2.4 \\% \\\\ \\hline
> > > MG-FNO &88 & 16 & 40 & 1.8 \\% \\\\ \\hline
> > > Tucker MG-TFNO &80& 16 & 46 & 1.3 \\% \\\\ \\hline
> > > \\end{array}$
> > >
> > > We optimized the best possible matching neural operator that fit in memory for a V100 GPU. This is the width 12 model in the table. We then compare its performance with the multigrid approach with a neural operator as large as fits into the same V100 GPUs i.e. each width in the table has been optimized to be as large as memory allows.  We will add those details to the manuscript, alongside with the new results.
> > >
> > > Finally, we show that our method enables larger models, by reducing memory usage. Again, we would like to refer to our response to reviewer J6V8.
> > >
> > > -------
> > > We hope our clarifications, which we will add to the manuscript, satisfactorily address reviewer zgdt’s comment.
> > >
> > >
> > > [1] https://arxiv.org/pdf/2203.13181.pdf
> > >
> > > [2] https://arxiv.org/pdf/2210.01074.pdf

---

### Official Review · Reviewer_u8oi · 2022-10-21

**Confidence:** 4
**Correctness:** 3
**Technical Novelty And Significance:** 3
**Empirical Novelty And Significance:** 3
**Recommendation:** 5

**Clarity, Quality, Novelty And Reproducibility:**

The paper is written clearly, up to the experiment section. There very interesting details are missing, which make it really hard to judge the impact. In the current version, the experiments are not reproducible

**Strength And Weaknesses:**

Strengths:
-	The paper presents a very interesting approach to a well-known problem: how to scale neural PDE surrogates efficiently to large resolutions
-	Writing the FNO basic block as single parameter tensor, and then applying matrix factorization on that is really neat. I think the idea has great potential.
-	The multi-grid domain decomposition makes a lot of sense too.


Weakness:
-	The biggest concern are the experiments. I cannot find any information on the grid, and any ablation on the baseline method. I could very well be that FNO with much less modes would perform equally well, which would make the achieved compression rates obsolete. It is really hard to judge the results without knowing the grid sizes and having some visual image of the data. Looking at Figure 3, the problems however don’t really seem to have a very high resolution. What is the temporal resolution? Which solver is used to obtain the data? What are the boundary conditions, stating Reynolds numbers and not the boundary condition is problematic.
-	There is no mentioning of other operator learning methods (e.g. DeepONet from Lu et al.) as well as a discussion of many other neural PDE surrogate methods. There is a huge body of neural PDE solvers/surrogates and omitting 95% of that is not good practice. I however admit that not everything is relevant for this paper. DeepOnet, Adaptive Fourier Neural Operators, multigrid methods, Vision Transformers are.
-	Connecting to the previous point: I am also very curious how this relates to Vision Transformers or Adaptive Fourier Neural Operators. Also recently UNets get tremendous performances in high-resolution image generation tasks, how are they doing, or even more interestingly can this approach be applied to e.g. UNets/DeepONets as well?
-	What do I see in Figure 5 right? What is H1 loss there? Both y axes state L2 loss? That fact that there is such a gap between FNO train vs test can very well be due to the rather basic training scheme (no warmup, lr annealing can surely be improved) or can be a sign that better regularization for the baseline is needed.

Lu, Lu, Pengzhan Jin, and George Em Karniadakis. "Deeponet: Learning nonlinear operators for identifying differential equations based on the universal approximation theorem of operators." arXiv preprint arXiv:1910.03193 (2019).


**Summary Of The Paper:**

The paper proposes a neural operator that scales to large resolutions by leveraging local and global structures through decomposition of both the input domain and the operator’s parameter space.
This is achieved by writing an FNO operation with a single tensor, and subsequently using low-rank tensor factorization.


**Summary Of The Review:**

The paper presents a strong idea, which is also practically very relevant. I highly encourage the authors to work on the experiment section and to work out the benefits of the methods as well as relation to other methods in more detail.

---

> ### Author Response · Authors · 2022-11-20
> **Response to Reviewer U8OI (1/2)**
>
> We would like to thank reviewer u8oi for the detailed comments
>
> > *The biggest concern are the experiments.*
>
> We thank the reviewer for the feedback. We will add all missing experiment details to the updated manuscript. We answer each specific point below.
>
> > *I cannot find any information on the grid, and any ablation on the baseline method.*
>
> We chose the best model that we could fit into a GPU for the baseline FNO and carefully validated all the hyper-parameters. In addition, to allow for fair comparison, we use the exact same setting and hyperparameters for our TFNO, where we simply additionally varied the compression ratio. For details on the grid, please see the responses to review j6v8.
> > *I could very well be that FNO with much less modes would perform equally well, which would make the achieved compression rates obsolete.*
>
> In Fig. 4.a, we perform a comparison of the TFNO and an FNO with the same number of parameters (trimmed FNO), for various compression ratios, from 0 to 400. This was achieved by reducing the width of the network, which was shown in (https://arxiv.org/abs/2203.13181) to be the most important parameter which is why we focused on that experiment. In addition, the width has the most impact on the total amount of parameters and reducing the number of modes results in limited compression.
> However, we ran additional experiments to empirically validate this (additional experiment 3).
>
> As we can observe, compression by reducing the modes rather than the width results in worse performance. In addition, for equivalent compression ratios, a compressed FNO is always worse than MG-TFNO which has lower error even for compression ratios > 100x. Note that lower compressions in MG-TFNO also leads to better performance as the latent low-rank tensor factorization helps reduce over-fitting.
>
> $\\begin{array} {|r|r|r|r|r|r|}\\hline
> Model & CR  &modes-width  &modes-height & model-width & l2-error\\\\ \\hline
> FNO & 23x & 7 & 15 & 40 & 0.032 \\\\ \\hline
> FNO & 10x & 7 & 15 & 64 & 0.030 \\\\ \\hline
> FNO & 23x & 5 & 8  & 64 &   0.041 \\\\ \\hline
> TFNO & 172x & 24 & 40  & 64 &   0.0236 \\\\ \\hline
> \\end{array}$
>
>
> > *It is really hard to judge the results without knowing the grid sizes and having some visual image of the data. Looking at Figure 3, the problems however don’t really seem to have a very high resolution.*
>
> We thank the reviewer for this good suggestion, we will include the visualization of the data.
> For all ablations, we use resolutions of 128x128 to mitigate the high cost of computational time and resources. Due the resolution invariance of the FNO, the results hold for any resolution that is higher. We are adding further experiments at resolution 512x512 to further illustrate a use case for our approach.
>
> > *What is the temporal resolution? Which solver is used to obtain the data?*
>
> The operator learning task is to map the source function to the solution at the 5th second, so there is no temporal resolution. We used a solver 512x512 with an adaptive time step to generate the data, as described in the appendix, section A.2.
>
> > *What are the boundary conditions, stating Reynolds numbers and not the boundary condition is problematic.*
>
> For the case of the Navier Stokes equation, the domain is a two dimensional periodic torus, as detailed in Appendix A.2.
>
>
> > *There is no mentioning of other operator learning methods (e.g. DeepONet from Lu et al.) as well as a discussion of many other neural PDE surrogate methods. There is a huge body of neural PDE solvers/surrogates and omitting 95% of that is not good practice. I however admit that not everything is relevant for this paper. DeepOnet, Adaptive Fourier Neural Operators, multigrid methods, Vision Transformers are.*
>
> We thank the reviewer for pointing these out, we will add a discussion about DeepONet, adaptive FNO, multi-pole, and vision transformers. Please note that our method is invariant to the choice of neural operator, for that reason we only focused on the state-of-the-art method.
>
>
> > *Connecting to the previous point: I am also very curious how this relates to Vision Transformers or Adaptive Fourier Neural Operators.*
>
> Relation to vision transformers and adaptive FNO. Both of the mentioned models learn maps from the input to the output while multi-grid maps the input at different scales to the output on the corresponding domain. We will incorporate a further discussion into the paper.
>
>
> > *Also recently UNets get tremendous performances in high-resolution image generation tasks, how are they doing, or even more interestingly can this approach be applied to e.g. UNets/DeepONets as well*
>
> The idea of multi-grid is universal, it can be used for both neural networks and neural operators. However, in this paper, we are concerned with operator learning tasks. We seek to train neural operators that are mapped between function spaces. Please note that neither U-net nor DeepONet fit the setting of this paper.

---

> > ### Author Response · Authors · 2022-11-20
> > **Response to Reviewer U8OI (2/2)**
> >
> >
> > > *What do I see in Figure 5 right? What is H1 loss there? Both y axes state L2 loss?*
> >
> > Thank you for pointing this out, indeed, in Figure 5, the y-axis on the right figure should state H1 loss, we will update this in the updated manuscript.
> >
> > > *That fact that there is such a gap between FNO train vs test can very well be due to the rather basic training scheme (no warmup, lr annealing can surely be improved) or can be a sign that better regularization for the baseline is needed.*
> >
> > We used the same training scheme for both the baseline and our method. In particular, the learning rate and annealing were carefully tuned, further optimizing on (Li et al, 2020a). We used a weight decay of 1e-4, corresponding to an l2 regularization on the weights.
> > In addition, tensorization has already been shown in computer vision to be useful to reduce over-fitting, see (Yannis Panagakis et al, 2021).
> >
> > > *In the current version, the experiments are not reproducible*
> >
> > Please note that, in addition to adding all missing details to the manuscript, we will release code to reproduce all the experiments upon acceptance of the paper.

---

> > > ### Comment · Reviewer_u8oi · 2022-11-24
> > > **Acknowledging the rebuttal**
> > >
> > > I'd like to thank the authors for the detailed comments, and explanations. However, I can't see an updated paper version which allows me to check if the promises have been kept. Therefore at this point, I keep my score.

---

> > > > ### Author Response · Authors · 2022-11-29
> > > > **Response to reviewer u8oi**
> > > >
> > > > > *I'd like to thank the authors for the detailed comments, and explanations. However, I can't see an updated paper version which allows me to check if the promises have been kept. Therefore at this point, I keep my score*
> > > >
> > > > We are glad to hear that the reviewer is satisfied with the detailed comments and explanations as we put significant effort in those. We promise to update the paper with all these new details. As mentioned to other reviewers, this is standard for any conference review process  and the whole academic system hinges on academic integrity and we are surprised to see it questioned here. The reviews and answers are both public so it will be clear for everyone whether this is done or not.

---

### Official Review · Reviewer_J6V8 · 2022-10-24

**Confidence:** 4
**Correctness:** 3
**Technical Novelty And Significance:** 3
**Empirical Novelty And Significance:** 3
**Recommendation:** 5

**Clarity, Quality, Novelty And Reproducibility:**

Many details are missing from the paper including the resolution of data used in the experiments. But many details can be more clear from the code and data release. The paper otherwise is easy to follow. The idea of using lower ranked factorization with alternative decomposition is common but specific applications to FFT based architectures is less explored and novel.

**Strength And Weaknesses:**

**Strengths**

Given GPU memory bottlenecks, different compute-memory tradeoff methods need to be explored as we keep increasing data sizes in deep learning. FNO based methods scale even worse with respect to memory when number of tracked modes increases. Proposed approach based on tensor decomposition methods makes sense for lower rank representation of model weights.

**Weaknesses**

Whether this is _hardware-efficient_ is not explored in this work. Something like Monarch [1] is the appropriate comparison for modern hardware. Matching derivatives during training makes the method even more expensive. Some timing comparisons would be useful.
Using % for reporting test error makes it harder to understand whether the underlying errors should be considered good. Similarly it's unclear what is "high" resolution in the paper because the experiments don't mention the spatial or temporal resolution. Moreover it would be useful to actually clarify if an external PDE solver was used or if the solver was hand written for datagen.

Some of the claims about "deep learning models being orders of magnitude faster than conventional solvers" are based on incorrect comparisons; running on GPUs vs. CPUs, not using state-of-the-art PDE solvers rather comparing against hand written solvers, not taking into account approximation errors etc.

It's also a bit surprising that older neural operator methods like DeepONets [2, 3] are missing from related works.

[1] https://arxiv.org/abs/2204.00595
[2] https://arxiv.org/abs/1910.03193
[3] https://arxiv.org/abs/2111.05512


**Summary Of The Paper:**

The paper focuses on the problem of neural surrogates of high-resolution PDEs. To this end, it applies the principles of multi-grid domain decomposition to exploit the local structure in such data as well applies tensor factorization with low-rank regularaization to reduce the number of parameters. The method is evaluated on vorticity form of Navier-Stokes equations and shown to improve performance. Although it's unclear what was the resolution of data used in experiments.

**Summary Of The Review:**

There are important details missing from the experiments in the paper to make a clear judgement but the idea of using lower ranked tensor factorization and domain decomposition  makes sense for the PDE domain.

---

> ### Author Response · Authors · 2022-11-20
> **Response to reviewer J6V8 (1/2)**
>
> We would like to thank reviewer j6v8 for the detailed feedback.
>
> > *Whether this is hardware-efficient is not explored in this work. Something like Monarch [1] is the appropriate comparison for modern hardware. Matching derivatives during training makes the method even more expensive. Some timing comparisons would be useful.*
>
> We thank the reviewer for the feedback and we are running additional experiments which we describe below, to empirically answer this question.
> The main focus of our paper is to enable better models that would not typically be trainable without it. Specifically, we propose a decomposition in the input domain which enables embarrassingly parallel training of very large models and tensorization for decomposition of the parameter space which enables training those large models more efficiently, while mitigating overfitting.
>
>
> In terms of hardware efficiency, this translates in 2 ways:
> 1. we can train much larger models by distributing over patches and still train efficiently thanks to the multi-grid patching
> 2. by efficiently performing training directly using the tensorized weights, we can further reduce memory requirements.
>
> We ran experiments for both points, by training on larger resolution (512x512 discretization) and using the largest FNO and TFNO that fits in memory.  We are including additional experiments at 512x512 to demonstrate our method’s usage in enabling larger models and faster run-times by parallelization.
>
> $\\begin{array} {|r|r|r|r|r|}\\hline
> Model & Width & Patches & Padding & Relative L^2 Error  \\\\ \\hline
> FNO & 12 & 0 & 0 & 6.1 \\% \\\\ \\hline
> MG-FNO & 42 & 4 & 70 & 2.9 \\% \\\\ \\hline
> MG-FNO &66 & 4 & 53 & 2.4 \\% \\\\ \\hline
> MG-FNO &88 & 16 & 40 & 1.8 \\% \\\\ \\hline
> Tucker MG-TFNO &80& 16 & 46 & 1.3 \\% \\\\ \\hline
> \\end{array}$
>
> We optimized the best possible matching neural operator that fit in memory for a V100 GPU. This is the width 12 model in the table. We then compare its performance with the multigrid approach with a neural operator as large as fits into the same V100 GPUs i.e. each width in the table has been optimized to be as large as memory allows.  We will add those details to the manuscript, alongside with the new results.
>
> In addition, the tensorized formulation enables an efficient implementation obtained by directly contracting the activations with the factors of the decomposition, as detailed in section A.3 of the appendix, allowing us to further scale the size of the models we can fit in the memory. We ran some additional experiments to measure the memory usage of a regular FNO compared to the same TFNO. Specifically, we created an FNO keeping 32 modes for the width and 64 modes for the height and we benchmarked two network width (64 and 128), for a batch-size of 16. In both cases, we measured the memory usage of both a regular FNO and our proposed Tucker TFNO, on a single RTX 3080 GPU, for a single forward-backward pass, using [1] pynvml (pynvml.nvmlDeviceGetMemoryInfo) and [2] PyTorch (torch.cuda.max_memory_allocated). As can be seen, our model not only consumes less memory, it scales better as we increase the network’s size, without impacting runtime.
>
> $\\begin{array} {|r|r|r|r|r|r|}\\hline
> Model & width & Compression & GPU memory [1]  & GPU memory [2] &  runtime\\\\ \\hline
> TFNO (Tucker) & 64 & 79x & 5.66 GB & 1.74 GB & 1.95s\\\\ \\hline
> FNO & 64 & 1x & 5.88 GB & 2.01 GB & 2.03s\\\\ \\hline
> TFNO (Tucker) & 128 & 77x & 6.96 GB & 3.25 GB & 2.01s\\\\ \\hline
> FNO & 128 & 1x & 11.14 GB & 5.59 GB & 2.04s\\\\ \\hline
> \\end{array}$
>
> Lastly, regarding the approach proposed in [1] is about the deployment of butterfly structures of sparse operations to build efficient neural networks. The mentioned work in [1] is concerned with neural networks. It is not resolution invariant in the input and does not output a function. Therefore, it is not a map between function spaces. This is in contrast to neural operators that are resolution invariant, and output functions (Li et al. 2020a). The present paper is concerned with operator learning tasks and maps between function spaces.
>
> > *Using % for reporting test error makes it harder to understand whether the underlying errors should be considered good.*
>
> The reported numbers are relative L2 errors in the prediction, illustrated in percentage. The relative L2 allows us to express the magnitude of error relative to the magnitude of the signal. This is a standard metric used prevalently in works which apply machine learning to PDEs as it allows one to easily gauge the quality of the approximation.

---

> > ### Author Response · Authors · 2022-11-20
> > **Response to Reviewer J6V8 (2/2)**
> >
> >
> > > *Similarly it's unclear what is "high" resolution in the paper because the experiments don't mention the spatial or temporal resolution.*
> >
> > We thank the reviewer for this question. The operator learning setting for Navier-Stokes is about a map from a 2D source function to the solution at the second 5. The problem is fully described in the appendix. For our ablation studies, we used the spatial resolution of 128x128 for both input and output functions in order to maintain a reasonable training time for the vast number of experiments. Since the FNO has been shown to be resolutionally invariant, our results hold at any larger resolution for the input and output.
> >
> > We also included in the previous comment additional experiments at 512x512 to demonstrate our method’s usage in enabling larger models and faster run-times by parallelization.
> >
> > > *Moreover it would be useful to actually clarify if an external PDE solver was used or if the solver was hand written for datagen.*
> >
> > We used our own pseudo-spectral solver implemented in PyTorch. A reference to the details of the solver is given in the appendix. All data is obtained by solving the equations at a 512x512 spatial resolution with an adaptive time-step determined by the CFL condition.
> >
> > > *Some of the claims about "deep learning models being orders of magnitude faster than conventional solvers" are based on incorrect comparisons; running on GPUs vs. CPUs, not using state-of-the-art PDE solvers rather comparing against hand written solvers, not taking into account approximation errors etc.*
> >
> > We agree that this remark is too broad and have removed it from the manuscript. Please note, however, that in the framework of the present paper, the operator learning task is to map the input source function to the solution output at time step 5. For such a task, neural operators directly output the 5th-second solution while the solver has to solve the problem from time zero to time 5s with a small time-step as governed by the CFL condition. Under the assumption that FFTs are the bottleneck, our model executes (number of layers) x width FFTs which is usually $4*64 = 256$. On the other hand, a solver at a fixed time step of 0.001 (which is optimistic at high resolutions due to the delta_x^2 scaling in CFL from the viscous term), executing at least 2 FFTs per time step, must execute 10000 FFTs to reach time 5. Therefore, our neural operator is significantly faster even when compared to state-of-the-art solvers running on optimized hardware.
> >
> > > *It's also a bit surprising that older neural operator methods like DeepONets [2, 3] are missing from related works.*
> >
> > We thank the reviewer for the pointers to papers [2] and [3], which are maps from finite-dimensional spaces to function spaces to the paper. We will add them to the discussion.
> >
> > > *There are important details missing from the experiments in the paper to make a clear judgement but the idea of using lower ranked tensor factorization and domain decomposition makes sense for the PDE domain.*
> >
> > To this point we would like to thank the reviewers for the valuable suggestions. We will add the requested details of the empirical study to the paper, and comment on them in the code base when released, alongside with the paper.

---

> > ### Comment · Reviewer_J6V8 · 2022-11-24
> > **Thanks for your response**
> >
> > I don't see how given the current setup of the experiments [1] being a neural network vs yours being a "neural operator" is of any concern. You would need actual concrete examples of the things that [1] won't be able to approximate which is not how your problem has been setup.
> >
> > It's great that you are now evaluating on 512 $\times$ 512 resolution. However, I don't see the updated paper PDF yet, so will keep my score.

---

> > > ### Author Response · Authors · 2022-11-29
> > > **Response to reviewer J6V8**
> > >
> > > > *I don't see how given the current setup of the experiments [1] being a neural network vs yours being a "neural operator" is of any concern. You would need actual concrete examples of the things that [1] won't be able to approximate which is not how your problem has been setup.*
> > >
> > > In this work, we are concerned with learning neural operators that are, by definition, maps between function spaces. Prior works on neural operators, e.g., Li 2020a and Li2020b, establish the importance of learning discretization-invariant maps between function spaces. Such models can take input functions at any discretization and output functions that can be queried at any co-location point. The prior works illustrate how trained neural operators can be applied to a variety of discretizations.
> > >
> > > This distinction between regular neural networks and neural operators has already been established, discussed and studied, empirically and theoretically in the prior literature and is outside the scope of this paper. We build on that literature in this work, our proposed model is a neural operator and can be applied to any resolution. This is not a property possessed by neural networks, including their efficient implementation [1]. To further illustrate this point, we have incorporated the study for which we train our model on 512x512 while a regular NN using the same GPU memory as for 128, would have an exponential increase in the number of parameters as we increase the resolution.
> > >
> > > > *It's great that you are now evaluating on 512 × 512 resolution. However, I don't see the updated paper PDF yet, so will keep my score.*
> > >
> > > We are glad to see the reviewer appreciate the new experiment as we put significant effort in our answer, and would like to point out the other experiments we ran and detailed review we wrote and hope these are all considered when updating the rating. We cannot update the manuscript at present but promise to do so with all these additions if accepted. The whole academic system hinges on integrity and we are surprised to see it questioned here. The reviews and answers are both public so it will be clear for everyone whether this is done or not.

---

### Author Response · Authors · 2022-11-20
**General response to reviewers**

We would like to thank the reviewers for their thoughtful feedback, which will help improve the quality of our paper. We are glad to read that all reviewers found the motivation for our approach compelling and appreciate its novelty and that both reviewers J6V8 and u8oi found the paper clear.

We answered in detail to each comment individually. To better answer some of these comments, we have run additional experiments. We will add all these clarifications as well as the additional experiments to our updated manuscript.

**In summary**, we introduce a MG-TFNO which is a generic approach that can be used to augment any neural operator. It consists of 2 parts: a decomposition of the problem domain through multi-grid domain decomposition, and a decomposition of the parameter space of the operator through a joint latent tensorization.

Our method enables scaling up to larger models. This scale-up is normally impossible due to i) the sheer size of the models necessary to deal with large input sizes, which, especially when taking into account the activation sizes, cannot not fit in memory and ii) the difficulty of training exponentially larger models. We tackle both issues with a decomposition of the input domain to solve, which reduces the size of the model i) and a decomposition of the parameters to solve both i) and ii) by further reducing the memory requirements while enabling better training through low-rank regularization of the operator.

Note that instead of focusing on training larger models than currently possible, one could apply the method to make any model embarrassingly parallel and reduce processing time through data distributed training on multiple GPUs, while needed smaller models.

We demonstrate superior accuracy on high resolution data and show that our method works in settings where regular FNO does not fit in memory. Importantly, our method is generic and applicable to any operator.

**Additional experiments**
In addition to our point by point answer, we performed additional experiment to empirically support our answers:

1. Training very large models on 512x512, showing our approach scales to larger models and resolutions. We obtained a relative improvement of 5X over the best FNO model.

2. We benchmarked memory usage of both a regular FNO and our proposed TFNO and empirically confirm the scalability of our method, which enables larger models.

3. Compressing the baseline by reducing the number of modes in the Fourier modes rather than the width, and a combination of the 2. We show that i) reducing the width of the model is the best way to compress a regular FNO. We already performed a thorough comparison with such FNO with trimmed with in figure 4.a in the manuscript  and ii) the MG-TFNO largely outperforms all these compressed FNO models

---

### Comment · Area_Chair_j9SK · 2022-11-22
**Please respond as soon as possible if you still have questions on the paper.**

Please respond as soon as possible if you still have questions on the paper.

---

### Decision · Program_Chairs · 2023-01-20

**Decision:**

Reject

**Justification For Why Not Higher Score:**

NA

**Justification For Why Not Lower Score:**

NA

**Metareview: Summary, Strengths And Weaknesses:**

The focus of this paper is on using neural networks to predict solutions to partial differential equations (PDEs) locally and combining these predictions into a global solution. The proposed method, called the neural operator, is able to handle large resolutions by taking advantage of both local and global structures in the input domain and the operator's parameter space. It is also able to achieve better performance than previous methods with significantly fewer training samples, outperforming the FNO when trained with only half as many samples.

The reviewers raised concerns on various aspects of the paper, and expected the authors to upload a revised manuscript to address (or partially address their concerns). Unfortunately, the authors did not make such a revision, and the authors' rebuttal was insufficient for them to raise scores. Moreover, Reviewer zgdt has the concern on no significant improvement over existing methods.

**Summary Of Ac-Reviewer Meeting:**

NA